# MergeMoE: Efficient Compression of MoE Models via Expert Output Merging

## Abstract

The Mixture-of-Experts (MoE) technique has proven to be a promising solution to efficiently scale the model size, which has been widely applied in recent LLM advancements. However, the substantial memory overhead of MoE models has made their compression an important research direction. In this work, we provide a theoretical analysis of *expert merging*, a recently proposed technique for compressing MoE models. Rather than interpreting expert merging from the conventional perspective of parameter aggregation, we approach it from the perspective of merging experts' outputs. Our key insight is that the merging process can be interpreted as inserting additional matrices into the forward computation, which naturally leads to an optimization formulation. Building on this analysis, we introduce MergeMoE, a method that leverages mathematical optimization to construct the compression matrices. We evaluate MergeMoE on multiple MoE models and show that our algorithm consistently outperforms the baselines with the same compression ratios.

## 1 Introduction

Large Language Models (LLMs) (Brown et al., 2020; Ouyang et al., 2022; Chowdhery et al., 2023; Achiam et al., 2023) have demonstrated outstanding performance in a wide spectrum of natural language processing (NLP) tasks. The improvement in the performance of LLMs is due to the scaling parameters (Kaplan et al., 2020), which also brings a high computational cost. The Mixture-of-Experts (MoE) architecture (Jacobs et al., 1991; Shazeer et al., 2017; Fedus et al., 2022; Zhou et al., 2022) is proposed to control computational cost while scaling the model parameters. In the typical MoE design, the input tokens are routed to several number of experts, trading higher memory overhead for lower computational cost. Recent advancement in LLMs has widely applied the MoE architecture (Rajbhandari et al., 2022a; Liu et al., 2024; Team, 2024; Jiang et al., 2024; Shen et al., 2024; Wei et al., 2024; Yang et al., 2025), which shows its significant potential in LLM studies.

The large number of parameters in the MoE model also makes its deployment relatively difficult, especially when resources are limited. The research community has proposed different ways to reduce the LLM's demand for resource, such as quantization (Dettmers et al., 2022; Yao et al., 2022; Xiao et al., 2023), knowledge distillation (Hinton et al., 2015; Gou et al., 2021), low-rank decomposition (Yu et al., 2017) and model pruning (Singh & Alistarh, 2020; Fang et al., 2023; Theus et al., 2024). Muralidharan et al. (2024) further shows that compressing pretrained large language models with knowledge distillation can produce smaller, high-quality models at much lower training cost. In this paper, we study model compression for MoE models via expert merging. M-SMoE (Li et al., 2023) demonstrates the potential of clustering and merging experts to reduce model size, but its merging algorithm is heuristic in nature and lacks theoretical support. Based on a new analysis, we propose an improved merging strategy that provides better theoretical grounding and achieves superior performance.

We begin by analyzing the theoretical foundation of the expert merging for MoE models. Rather than viewing expert merging from the traditional perspective of merging experts' parameter, we approach it from the perspective of merging experts' outputs. Our key insight is that the merging process can be interpreted as inserting additional matrices into the forward computation, which naturally leads to an optimization formulation. This analysis explains both why the prior work on expert-merging is effective and why residual errors remain. Building on the insight, we propose

MergeMoE, a novel expert-merging algorithm that explicitly optimizes the associated matrices. We merge experts by weighted averaging, where the usage frequency serves as the weight; we further prove this weighting scheme is optimal. To determine the internal parameters of merged experts, We employ the least-squares method, which provides an effective and practical way to compute the compression matrices.

Our main contribution can be summarized as follows.

- In §3, we provide theoretical insights into expert merging for MoE models and discuss how prior work on expert merging aligns with our analysis.

- In §4, we introduce MergeMoE, a method motivated by these theoretical insights, which focuses on merging experts' outputs using mathematical tools.

- In §5, we present experimental evaluations of MergeMoE. The results demonstrate that Merge-MoE consistently outperforms the baselines at the same memory compression ratios.

We also discuss the limitations of our work and the usage of LLMs in Appendices A and B.

## 2 RELATED WORKS

**Mixture-of-Experts models.** The Mixture-of-Experts (MoE) models have become a prevalent approach, which enable efficient expansion of neural network capacity while keeping computational costs under control. Shazeer et al. (2017) introduces a Sparsely-Gated Mixture-of-Experts architecture within LSTM models, which effectively boosts the model's capacity and enhances performance on downstream tasks. Fedus et al. (2022) applies the idea in the transformers and proposes the Switch Transformer architecture. Rajbhandari et al. (2022a;b) adopt the shared experts in their MoE architecture. Many recent LLMs (Liu et al., 2024; Jiang et al., 2024; Shen et al., 2024; Wei et al., 2024; Yang et al., 2025) apply the MoE technique to efficiently scale up the model capacity.

**Model Compression.** As the scale of the the the models continues to increase, researchers have also started to explore how to compress the models, making them easier to deploy. Model pruning is a typical technique to compress the models. Wang et al. (2019) proposes a network reparameterization and structured pruning solution on Resnet and VGG model. Fang et al. (2023) analyzes the dependency graph in the network and presents a parameter pruning solution on various models architecture. Theus et al. (2024) incorporates the optimal transport technique and proposes Intra-Fusion for pruning. All these works are targeted at the general LLM architecture.

On the other hand, model compression for MoE models is not fully studied. M-SMoE (Li et al., 2023) first propose to merge experts in order to compress the MoE models. M-SMoE clusters experts into groups and merges those within each group by computing a weighted average of the corresponding weight matrices, where the weights are determined by the experts' usage frequencies. Xie et al. (2024) follows the previous pruning approaches in LLMs and ignores the unique features of MoE models. He et al. (2023) merges multiple experts into a single expert from a computational perspective, which does not reduce memory cost.

## 3 BACKGROUND AND THEORETICAL INSIGHTS

In this section, we first provide a brief overview of the MoE architecture. We then present theoretical insights into expert merging, which recast the merging process as introducing additional matrices in the forward computation and framing it as an optimization problem. Finally, we revisit prior expert-merging algorithm and show how they can be interpreted within our theoretical framework, thereby clarifying their limitations.

### 3.1 PRELIMINARY

We begin by introducing the MoE architecture. Let $N$ be the number of experts and $K$ be the number of activated experts per token. The MLP module consists of a router and $N$ experts, where the router has weight matrix $W_r$. Given an input $X$, the router computes $softmax(W_r X)$ and selects top-$K$ experts according to the highest scores. We denote the $i^{th}$ expert as $E_i$, which follows the SwiGLU

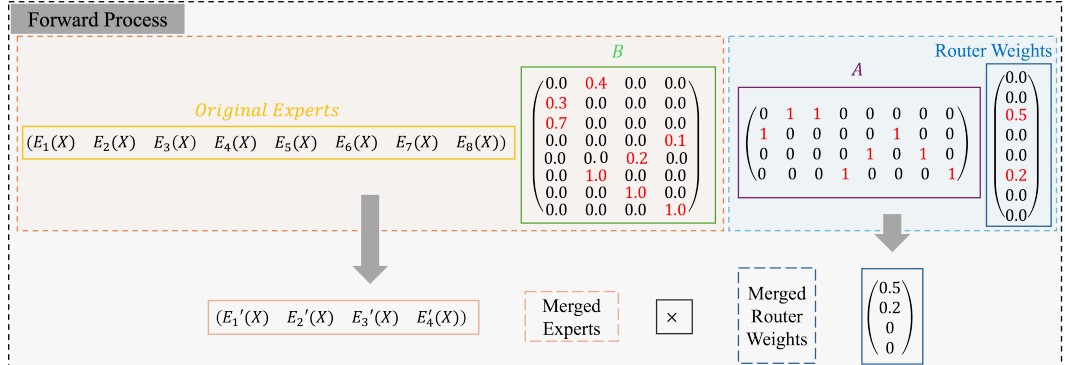

Figure 1: An overview of how the merging algorithm changes the forward process of the MoE module. It shows the transition from an initial 8-expert configuration (top-2 activation per token) to 4 experts after compresion.

design and contains three weight matrices $W_D$, $W_U$ and $W_G$ and a non-linear activation function $\sigma$. With a slight abuse of notation, we use $E_i(X)$ to denote its output on input $X$, which is given by:

$$E_i(X) = W_D(\sigma(W_G X) \odot (W_U X)),$$

where $\odot$ denotes the Hadamard product. After the selected $K$ experts compute their outputs, the final result is obtained as a weighted average of these outputs, with weights given by the corresponding top-$K$ entries of $softmax(W_r X)$. Formally, the forward computation can be written as

$$[E_1(X) \quad E_2(X) \quad \ldots \quad E_N(X)] \cdot mask\_top\_K(softmax(W_r X))^\top$$

Let

$$Y = [E_1(X) \quad E_2(X) \quad \ldots \quad E_N(X)],$$

then the formula above can be simplified as

$$Y \cdot mask\_top\_K(softmax(W_r X))^\top \tag{1}$$

Here $mask\_top\_K(\cdot)$ denotes the operator that sets all but the top-$K$ entries to zero. We emphasize that Eq 1 describes an equivalent computational view; in practice, masked experts are skipped and do not contribute to computation.

## 3.2 Insights for Expert Merging

We next consider merging experts within a single MoE layer, reducing the number of experts from $N$ to $M$. To achieve this, the experts are first clustered into $M$ groups, and the experts within each group are then merged to form a new expert. Traditionally, model pruning have focused on the parameter space. In this view, experts that are considered "similar" are grouped and merged by averaging or weighted averaging their parameters, under the intuition that combining similar parameters reduces approximation error. Routing weights for the merged experts are then computed as the sum of the original experts' routing weights. In contrast, we argue that experts merging should focus on merging the experts' outputs.

As shown in Figure 1, summing the routing weights of the merged experts is equivalent to multiplying by a summation matrix $A$, defined as:

$$A_{ij} = \begin{cases} 1, & \text{the original } j^{th} \text{ expert is classified into } i^{th} \text{ cluster} \\ 0, & \text{otherwise} \end{cases} \tag{2}$$

In Figure 1, the clustered groups are $(E_2, E_3), (E_1, E_6), (E_5, E_7), (E_4, E_8)$. Given original routing weights $(0, 0, 0.5, 0, 0, 0.2, 0, 0)^\top$, the weights after merging become $(0.5, 0.2, 0, 0)^\top$. Motivated by this observation, we shift the target of weighted averaging from experts' parameters to their outputs, which can be expressed as multiplication by a matrix $B$:

$$B_{ij} = \begin{cases} w_{ij}, & \text{if the original } i^{th} \text{ expert is assigned to the } j^{th} \text{ cluster with weight } w_{ij} \\ 0, & \text{otherwise} \end{cases}$$

Consequently, the forward pass can be rewritten as

$$Y \cdot B \cdot A \cdot mask\_top\_K(softmax(W_r X))^\top$$

This allows us to move from a previously qualitative view of parameters merging to a quantitative one, by formulating it as a linear optimization problem, where the objective is to choose $A$ and $B$ such that the merged forward output approximates the original MoE forward computation in Equation 1.

The remaining challenge is how to set the parameters of the merged experts such that their outputs approximate a linear combination of the original experts' outputs. Let $E_i'$ denote the $i^{th}$ merged expert. It should approximately satisfy

$$E_i'(X) = \sum_j B_{ji} E_j(X), \forall X.$$

For example, in Figure 1, the first group consists of the $2^{nd}$ and $3^{rd}$ experts, with weights 0.3 and 0.7, respectively. Then the merged expert $E_1'$ should approximately satisfy $E_1'(X) = 0.3E_2(X) + 0.7E_3(X), \forall X$.

We find that

$$E_i'(X) = \sum_j B_{ji} E_j(X) = \sum_j B_{ji} W_{Dj}(\sigma(W_{Gj}X) \odot (W_{Uj}X))$$

$$= [B_{1i}W_{D1}, B_{2i}W_{D2}, \cdots, B_{Ni}W_{DN}](\sigma(\begin{bmatrix} W_{G1} \\ W_{G2} \\ \vdots \\ W_{GN} \end{bmatrix} X) \odot (\begin{bmatrix} W_{U1} \\ W_{U2} \\ \vdots \\ W_{UN} \end{bmatrix} X))$$

If we set the parameters of merged experts as

$$W_{Di}' = [B_{1i}W_{D1}, B_{2i}W_{D2}, \cdots, B_{Ni}W_{DN}], W_{Gi}' = \begin{bmatrix} W_{G1} \\ W_{G2} \\ \vdots \\ W_{GN} \end{bmatrix}, W_{Ui}' = \begin{bmatrix} W_{U1} \\ W_{U2} \\ \vdots \\ W_{UN} \end{bmatrix},$$

then the merged experts $E_i'(X) = W_{Di}'(\sigma(W_{Gi}'X) \odot (W_{Ui}'X))$ can satisfy the requirement without incurring any approximation error. However, this construction only works because we allow the intermediate dimensions to grow with the number of merged experts. As a result, both the parameter size and the computational cost remain unchanged. To ensure that each merged expert has the same parameter scale as a standard expert, we need to reduce the intermediate dimensionality. We then introduce dimension reduction matrices $T_1, T_2, T_3$ and express the merged expert as

$$E_i'(X) = W_{Di}'T_1(\sigma(T_2 W_{Gi}'X) \odot (T_3 W_{Ui}'X)), \tag{3}$$

which transforms the problem into finding suitable $T_1, T_2, T_3$ to reduce the approximation error.

### 3.3 M-SMoE UNDER OUR OUTPUT-MERGING VIEW

The prior work on expert merging, M-SMoE, adapts the traditional view of merging parameter. M-SMoE merges experts in the same cluster by weighted averaging the parameters of each weight matrices, with usage frequencies as the weights. Under our output-merging view, it is equivalent to set $T_1, T_2, T_3$ as follows.

$$T_1 = \begin{bmatrix} I, \\ I, \\ \vdots \\ I \end{bmatrix}, T_2 = [B_{1i}I, B_{2i}I, \cdots B_{Ni}I], T_3 = [B_{1i}I, B_{2i}I, \cdots B_{Ni}I]. \tag{4}$$

The $T_1, T_2, T_3$ settings are not derived from quantitative optimization, and thus there remains room for further improvement.

## 4 METHODOLOGY

Finding the optimal $T_1, T_2, T_3$ that minimize the approximation error is challenging, because it contains a non-linear activation function and a Hadamard product. We propose a strategy that decouples the optimization of $T_1$ and $T_2, T_3$.

We first assume the $T_2$ and $T_3$ are fixed and focus on the $T_1$ alone. Given a sampled inputs $\hat{X}$, according to Equation 3, the $T_1$ should approximately satisfy

$$T_1(\sigma(T_2 W'_{Gi}\hat{X}) \odot (T_3 W'_{Ui}\hat{X})) = \sigma(W'_{Gi}\hat{X}) \odot (W'_{Ui}\hat{X}) \tag{5}$$

Because $T_2, T_3$ and input samples $\hat{X}$ are given, we can compute $P = (\sigma(T_2 W'_{Gi}\hat{X}) \odot (T_3 W'_{Ui}\hat{X}))$ and $Q = \sigma(W'_{Gi}\hat{X}) \odot (W'_{Ui}\hat{X})$ and reduces the problem to a linear system $T_1 P = Q$. Since this forms a linear least squares problem, $T_1$ admits a closed-form solution

$$T_1 = QP^\dagger, \tag{6}$$

where $P^\dagger$ denotes the Moore-Penrose pseudoinverse of $P$.

The $T_2$ and $T_3$ are closely associated with the non-linear activation function and the Hadamard product. This tight integration introduces intrinsic non-linearities that prevent the objective function from being reformulated as a linear optimization problem, thereby precluding the existence of a closed-form solution for their joint optimization. Therefore we let $T_2$ and $T_3$ represent weighted averages within clusters and set them according to Equation 4. To reduce the error caused by weighted average, when clustering the experts, we employ the similarity of the concatenated results of the matrix $W_U$ and the matrix $W_G$ of experts as the metric to measure the distance between two experts. Then weighted average is performed among experts with similar $W_U$ and $W_G$, and the approximation error can be reduced.

Once the clustering method is determined, the matrix $A$ is also uniquely fixed according to Equation 2. We use the relative usage frequency of the experts as the weight for the weighted average within the cluster. It is noticeable that M-SMoE also applies relative usage frequency as the weight. However, it selects this scheme primarily based on experimental performance, while we provide theoretical proof for its optimality.

Our aim is to minimize the error between the compressed output and the original output, which is the Frobenius norm of

$$(YBA - Y) \cdot mask\_top\_K(softmax(W_r X))^\top$$

We define a "Quasi-Frobenius" norm $QF(Y)$:

$$QF(Y) = [||E_1(X)||_F^2, ||E_2(X)||_F^2, ..., ||E_N(X)||_F^2] \in \mathbb{R}^N$$

We suppose that the router logits and the output of experts are independent. Consider taking a large number of samples, if the distribution of the frequency of expert usage is already known, explicitly, let the expected number of times the $i$-th expert is used be $f_i$, and denote $Y_0 = \mathbb{E}_{X \sim \pi} Y$, where $\pi$ is the distribution of the input $X$. Then the function $mask\_top\_K$ can be unpacked as an expected value, which leading to an simplified lower bound for the above equation:

$$\mathbb{E}_{X \sim \pi}[||(YBA - Y)mask\_top\_K(softmax(W_r X))^\top||_F^2]$$
$$= \mathbb{E}_{X \sim \pi}[(Y(BA - I_N))QF \cdot mask\_top\_K(softmax(W_r X))^\top]$$
$$= \mathbb{E}_{X \sim \pi}[Y((BA - I_N)QF] \times \mathbb{E}_{X \sim \pi}[mask\_top\_K(softmax(W_r X))^\top]$$
$$\geq Y_0((BA - I_N)QF) \times [f_1, f_2, ..., f_N]^\top$$

where $I_N$ denotes the identity matrix in $\mathbb{R}^{N \times N}$.

For a given clustering approach, each pre-merger expert should correspond to exactly one post-merger expert. Also, a post-merger expert is the weighted sum of its corresponding pre-merger ones. This is equivalent to each row of $A$ having exactly one 1 and the rest are 0, and each row of $B$ having non-zero values only at the indices of its cluster.

**Theorem 1.** *Given $A \in \mathbb{R}^{M \times N}$, $Y_0 \in \mathbb{R}^{K \times N}$, each column of $A$ has exactly one 1 and the rest are 0. Let $B \in \mathbb{R}^{N \times M}$, $v_1, v_2, ..., v_M$ be the columns of $B$. Let $C_i$ be the indices corresponding to*

*the non-zero values of the $i$-th column of $A$. For $i = 1, 2, ..., M$, $v_i$ has non-zero values only at the indices in $C_i$. Then:*

$$v_i[j] = \begin{cases} \frac{f_j}{\sum\limits_{k \in C_i} f_k}, & if \ j \in C_i \\ 0, & otherwise \end{cases}$$

*is a minimal point of the function:*

$$Y_0((BA - I_N)QF) \times [f_1, f_2, ..., f_N]^\top$$

For a detailed proof of the theorem, please refer to Appendix C.

**Summary of the algorithm design.** We have explained all the design choices in our algorithm. Our algorithm is summarized as follows.

1. Clustering. Experts with top-$M$ usage frequencies are selected as the clustering center, and the other experts are classified according to their distance from the experts in the clustering centers. We uses the similarity of the concatenated results of the matrix $W_U$ and the matrix $W_G$ as the metric for the distance between two experts.

2. Merging the experts within the same cluster. Within the cluster, we use the relative usage frequency of each expert as the weight. We set the compression matrix $T_2, T_3$ according to Equation 4, which represent the weighted average. Then we utilize input samples $\hat{X}$ and apply the least squares method according to Equation 6 to compute the closed form result of $T_1$. Finally $W_D' T_1, T_2 W_G', T_3 W_U'$ will be outputted as the weight matrices of the merged expert.

It is noticeable that our technique can also be applied to those MoE models with shared experts. In models with shared experts, the shared experts and routed experts are usually independent during the forward pass. Therefore, the routed experts can be directly compressed according to our algorithm.

## 5 EVALUATION

### 5.1 SETUP

**Models and Datasets.** We used three open-source MoE models for evaluation: DeepSeekMoE (Rajbhandari et al., 2022a), Qwen1.5-MoE-A2.7B (Team, 2024), and Qwen3-30B-A3B (Yang et al., 2025). We summarize the configurations of the three models in Appendix E.1. The experiments are conducted on seven NLP datasets: MRPC (Dolan & Brockett, 2005) for paraphrase identification, WinoGrande (Sakaguchi et al., 2021) for coreference resolution, SQuAD (Rajpurkar et al., 2016) for extractive QA, Hellaswag (Zellers et al., 2019) for commonsense reasoning, PIQA (Bisk et al., 2020) for physical interaction reasoning, ARC easy and ARC challenge (Clark et al., 2018) for scientific reasoning. In Appendix E.3 we further evaluate the performance of MergeMoE on the instruction following benchmark IFEval (Zhou et al., 2023).

**Evaluation Details.** The merging algorithms are conducted on a single NVIDIA H20 with 96GB memory, and the evaluation is conducted on two NVIDIA H20. We use DCLM (Li et al., 2024) to evaluate the performance of models in downstream tasks. We use M-SMoE Li et al. (2023) as the main baseline for the comparative experiments. Considering the lack of work on experts merging, we also uses the baselines in the experiments of the M-SMoE, which adapt Average (Choshen et al., 2022) and ZipIt (Stoica et al., 2023) in the expert merging scenarios. In the comparative experiments, we ensure that both our solution and the baselines merge the same set of layers, and the compression ratios are also the same. For the M-SMoE, although it describes a way to adjust the compression ratios of each layer, we found in our evaluations that it may lead to much worse results. Therefore, we simply fix the compression ratios for all layers to be consistent, and we believe it is still a fair setting.

### 5.2 PERFORMANCE OF MERGEMOE

We compare the performance of MergeMoE with baseline algorithms on three MoE models. For the evaluation on the Qwen3-30B-A3B model, we additionally use Qwen3-4B as a dense baseline,

Table 1: Performance evaluation of MergeMoE and the baselines on the Qwen3 model.

| Strategies | Model Size | WinoGrande | ARC easy | ARC challenge | Hellaswag | PIQA | SQuAD | MRPC |
|---|---|---|---|---|---|---|---|---|
| Full | 30B | 74.27 | 84.89 | 67.49 | 76.38 | 81.72 | 66.61 | 72.55 |
| Dense | 4B | 67.96 | 81.31 | 60.07 | 68.21 | 77.37 | 64.22 | 75.74 |
| Average | 25B | 73.24 | 82.74 | 51.96 | 71.36 | 74.65 | 63.94 | 72.55 |
| ZipIt | 25B | 72.77 | 77.78 | 56.40 | 72.61 | 76.50 | 63.81 | 72.55 |
| M-SMoE | 25B | 73.95 | 82.87 | 61.77 | 74.12 | 80.79 | 64.28 | 72.30 |
| MergeMoE | 25B | 73.72 | 83.04 | 63.48 | 74.93 | 81.34 | 64.56 | 72.55 |

Table 2: Performance evaluation of MergeMoE and the baselines on the Qwen1.5 model.

| Strategies | Model Size | WinoGrande | ARC easy | ARC challenge | Hellaswag | PIQA | SQuAD | MRPC |
|---|---|---|---|---|---|---|---|---|
| Full | 14B | 72.30 | 76.98 | 50.60 | 77.14 | 80.79 | 60.36 | 72.06 |
| Dense | 4B | 66.85 | 72.55 | 42.75 | 70.00 | 77.97 | 60.54 | 62.99 |
| Dense | 1.8B | 61.25 | 65.07 | 35.49 | 60.14 | 74.32 | 49.53 | 68.87 |
| Average | 10B | 68.11 | 69.28 | 41.30 | 67.92 | 78.94 | 53.85 | 72.30 |
| ZipIt | 10B | 69.14 | 69.53 | 41.81 | 68.06 | 77.80 | 55.75 | 72.06 |
| M-SMoE | 10B | 68.98 | 71.00 | 41.55 | 68.87 | 79.27 | 54.99 | 72.30 |
| MergeMoE | 10B | 70.48 | 71.25 | 42.06 | 71.58 | 79.27 | 56.40 | 74.75 |

since among the Qwen-3 series it has the closest number of activated parameters to Qwen3-30B-A3B. For the evaluation on the Qwen1.5-MoE-A2.7B, we use Qwen1.5-1.8B and Qwen1.5-4B as dense baselines. For each model, we select a set of layers and a compression ratio; for each selected layer, the number of experts is reduced according to this ratio. All merging algorithms then merge the experts for these layers and evaluate the resulting performance. We also ensure the number of input samples is the same for all merging algorithms applied to the same model and dataset combination. The detailed hyper-parameter configurations, including the merging layers, compression ratios, and the number of input samples are described in E.2. For clarity, the highest-performing scheme is highlighted in blue, and the second-highest in yellow.

**Comparison on the Qwen3.** The experiment results are shown in Table 1. First, MergeMoE achieves the best performance on all tasks except the WinoGrande. On the WinoGrande task, the performance of MergeMoE is the second-highest, with only a $0.23$ gap from the best score. Second, the performance gap between MergeMoE and the full model is minimal. On the WinoGrande, PIQA and MRPC tasks, the performance drop compared to the full model is even less than $0.6$. Third, our solution significantly outperforms the dense model on most tasks. Notably, while the compressed model uses only 3B active parameters compared to 4B in the dense model, it still achieves superior performance, demonstrating the efficiency and effectiveness of our approach.

**Comparison on the Qwen1.5.** The experiment results are shown in Table 2. MergeMoE achieves the best performance on all tasks. Compared with the SOTA solution, M-SMoE, MergeMoE improves $1.5$ on the WinoGrande task, $2.71$ on the PIQA task, $1.41$ on the SQuAD task, and $2.45$ on the MRPC task. We also find that, MergeMoE significantly outperforms the Qwen1.5-1.8B dense model. Compared with Qwen1.5-4B dense model, it achieves better performance on WinoGrande, Hellaswag, PIQA, and MRPC tasks, and comparable performance on the others. As the compressed model has 2.7B active parameters, we believe our solution is efficient on the Qwen1.5 model.

**Comparison on the DeepSeekMoE.** The experiment results are shown in Table 3. Overall, MergeMoE achieves the best performance compared with baselines. Compared to M-SMoE, our approach achieves an improvement of $1.13$ on ARC easy and $1.16$ on Hellaswag. Compared to Average, MergeMoE achieves an improvement of $1.31$ on ARC easy and $1.2$ on ARC chanllenge. Compared to ZipIt, MergeMoE achieves an improvement of $2.71$ on Hellaswag. Besides, compared with the full model, the performance drop is negligible on most tasks.

**Summary.** We obtain the following observations from the experiment results. First, MergeMoE generally achieves the best performance among all the baseline algorithms. On all the three models, MergeMoE attains a improvement for most tasks. Second, the performance drop caused by compression is negligible in most cases. Third, MergeMoE outperforms the dense model with a

Table 3: Performance evaluation of MergeMoE and the baselines on the DeepSeekMoE model.

| Strategies | Model Size | WinoGrande | ARC easy | ARC challenge | Hellaswag | PIQA | SQuAD | MRPC |
|---|---|---|---|---|---|---|---|---|
| Full | 16B | 74.59 | 78.17 | 50.26 | 77.10 | 80.30 | 53.87 | 60.05 |
| Average | 12B | 73.48 | 74.53 | 45.90 | 75.53 | 79.81 | 54.17 | 60.54 |
| ZipIt | 12B | 73.09 | 75.55 | 47.53 | 72.61 | 79.00 | 54.65 | 60.54 |
| M-SMoE | 12B | 73.32 | 74.71 | 47.27 | 74.16 | 79.05 | 55.11 | 60.29 |
| MergeMoE | 12B | 73.64 | 75.84 | 47.10 | 75.32 | 79.87 | 54.27 | 60.78 |

Table 4: Evaluation of the cross-dataset generalization abilities for MergeMoE on the Qwen1.5 model. "Self-Sourced Samples" indicates using corresponding samples for each tasks, which follows the same setting in Table 2. The rest three rows use WinoGrande/ARC easy/Hellaswag for merging and evaluate on all tasks. To ensure fairness, we set the total number of sample tokens to be identical to 16K.

| Source of Input Samples | WinoGrande | ARC easy | ARC challenge | Hellaswag | PIQA | SQuAD |
|---|---|---|---|---|---|---|
| Self-Sourced Samples | 70.48 | 71.25 | 42.06 | 71.58 | 79.27 | 56.40 |
| WinoGrande | 70.40 | 67.72 | 43.69 | 70.11 | 77.86 | 54.33 |
| ARC easy | 68.58 | 72.47 | 42.32 | 67.94 | 76.99 | 54.60 |
| Hellaswag | 69.14 | 70.41 | 43.09 | 71.56 | 78.56 | 54.29 |

comparable number of active parameters. The results show that, MergeMoE effectively mitigates performance degradation from MoE model compression and demonstrates superior effectiveness.

## 5.3 EXTRA EXPERIMENTS

**Experiments on time cost.** We compare the time costs of MergeMoE and M-SMoE during the merging process, with results reported in Figure 3. Experiments are conducted on the WinoGrande task using the Qwen 1.5 model. In our setting, MergeMoE is run with a batch size of 128 input samples, and for each layer the number of experts is reduced from 60 to 30. Although MergeMoE is slower than M-SMoE, which is an expected outcome given its more complex operations, both methods complete within a minute. This makes the overall cost negligible. Moreover, since our merging algorithm runs efficiently on a single GPU, Merge-MoE imposes relatively low resource requirements.

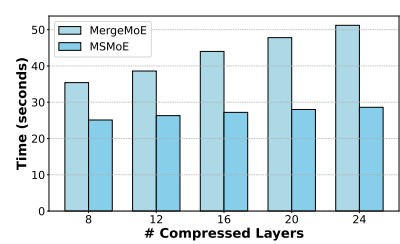

Figure 3: Comparison of the time cost.

**Experiments on different compression ratios.** We evaluate how different compression ratios affect the performance of models merged by our algorithm. The experiment is conducted on the WinoGrande task with Qwen 1.5 model. Two factors determine the compression ratio: (1) the number of layers involved in the merging process, and (2) the reduced number of experts in each merged layer. In Figure 2a we fix the number of compressed layers to 14 and vary the number of

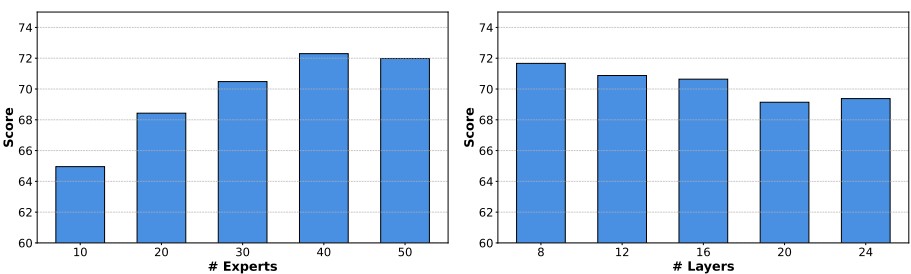

(a) Impacts of the number of reduced experts.  (b) Impacts of the number of compressed layers.

Figure 2: Experiments on the effects of different compression ratios.

Table 5: Ablation experiments on the compression errors.

| Strategies | WinoGrande | ARC easy | ARC challenge | Hellaswag | PIQA |
|---|---|---|---|---|---|
| Full | 72.30 | 76.98 | 50.60 | 77.14 | 80.79 |
| w/o merging errors | 71.27 | 73.11 | 43.69 | 72.91 | 79.60 |
| w/ merging errors | 70.48 | 71.25 | 42.06 | 71.58 | 79.27 |

reduced experts. In Figure 2b we instead fix the number of reduced experts to 30 and vary the number of compressed layers. Experimental results indicate that the model accuracy gradually decreases as the compression ratio increases. Furthermore, comparing the impacts of reducing expert count versus increasing compressed layers, we find the former has a more significant effect. This suggests that when implementing the compression algorithm, we should avoid excessive compression of the number of experts in a single layer and instead expand the number of compressed layers.

**Experiments on the number of input samples.**
MergeMoE relies on input samples to apply least-squares method for computing an accurate compression matrix $T_1$, and its performance is directly affected by the number of such samples. We evaluate this effect using the Qwen 1.5 model on the WinoGrande task, and the configuration of the compression layers and the compression ratios are the same with the experiment in Table 2. As shown in Figure 4, MergeMoE fails completely when the sample size falls below a critical threshold (32 in our experiment). Since WinoGrande is a binary-choice dataset, scores around $50\%$ correspond to random guessing. In contrast, once the sample size exceeds the threshold (36), performance improves rapidly and then continues to in-

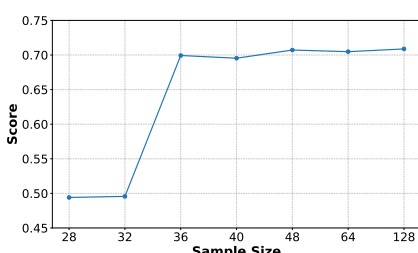

Figure 4: Evaluation on the impact of the number of sample size.

crease more gradually with additional samples. The results indicate that, MergeMoE is sensitive to sample size. Our algorithm achieves reliable performance only when the number of input samples exceeds this critical threshold. Moreover, increasing the number of samples beyond the threshold consistently leads to further performance gains.

**Cross-dataset generalization.** We explore the ability for the MergeMoE to generalize across different datasets. Specifically, we apply MergeMoE using input samples sourced from a single dataset, then evaluate the resulting compressed model across all tasks. As shown in Table 4, the model merged from a single source dataset achieves scores only slightly lower than those from models merged with self-sourced samples (i.e., samples taken from each respective benchmark). This indicates that our algorithm has cross-dataset generalization capability.

**Ablation on the compression errors.** As analyzed in 3.2, compression errors stem from clustering $(A, B)$ and expert merging $(T_1, T_2, T_3)$. To isolate their effects, we conduct an ablation experiment where clustering is retained but expert outputs are directly merged, thereby removing merging errors. As shown in Table 5, this variant outperforms the standard merging scheme, which is consistent with our analysis. The small performance gap further demonstrates the effectiveness of our least-squares method in mitigating merging errors.

## 6 CONCLUSION

In this paper we study how to compress MoE models by merging experts. We first analyze the theoretical essence of the expert merging in MoE models. Unlike the traditional view that focuses on merging expert parameters, we introduce a novel perspective that interprets expert merging as expert output merging. Under this perspective, the merging process can be formulated as inserting additional matrices into the forward computation. Building on this theoretical insight, we propose our solution, MergeMoE, which uses mathematical tools to optimize the design of the compression matrices in the expert-merging process. Our experiment results show that, compared with baseline algorithms, MergeMoE consistently achieves better performance at the same compression ratio.

## ETHICS STATEMENT

Our work focuses on algorithm design for compressing MoE models through expert merging. While the proposed method improves efficiency, it does not address or mitigate the biases present in the underlying training data. In particular, merging strategies that rely on usage frequency may disproportionately compress rarely activated experts, potentially degrading performance on minority or long-tail cases. Moreover, compressed models inherit the risks of their uncompressed counterparts, including the possibility of generating biased or misleading content.

## REPRODUCIBILITY STATEMENT

Our code is open-sourced at `https://anonymous.4open.science/r/MergeMoE_opensource-5DCB`. We further provide discussions on the implementation details of our solution in Appendix D.

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

## A  LIMITATION

The primary limitation of our method is its reliance on input samples. Our experiment on the number of input samples shows that, MergeMoE can only achieve strong performance when provided with a sufficient number of samples. The generalization experiment further indicates that merging on one dataset can transfer reasonably well to other tasks, but still underperforms compared to sampling directly from the corresponding benchmarks. We believe that diverse and high-quality input samples are essential for MergeMoE to reach its full potential.

## B  LLM USAGE STATEMENT

For this paper, we utilize LLMs as language-assistance tools to refine expression, such as enhancing grammatical accuracy and readability. All research concepts, theoretical analysis and methodologies are original contributions of the authors.

## C  THEORETICAL ANALYSIS OF THE MERGING WEIGHTS

**Theorem 1.** *Given $A \in \mathbb{R}^{M \times N}$, $Y_0 \in \mathbb{R}^{K \times N}$, each column of $A$ has exactly one $1$ and the rest are $0$. Let $B \in \mathbb{R}^{N \times M}$, $v_1, v_2, ..., v_M$ be the columns of $B$. Let $C_i$ be the indices corresponding to the non-zero values of the $i$-th column of $A$. For $i = 1, 2, ..., M$, $v_i$ has non-zero values only at the indices in $C_i$. Then:*

$$v_i[j] = \begin{cases} \dfrac{f_j}{\sum\limits_{k \in C_i} f_k}, & if \ j \in C_i \\ 0, & otherwise \end{cases}$$

*is a minimal point of the function:*

$$Y_0((BA - I_N)QF) \times [f_1, f_2, ..., f_N]^\top$$

*Proof.* Suppose that $a_1, a_2, ..., a_N$ are the column vectors of $A$, $v_1, v_2, ...v_M$ are the column vectors of $B$, $u_1, u_2, ...u_N$ are the column vectors of $BA$. Then

$$u_i = B \times a_i = \sum_{j=1}^{M} v_j \times a_i[j]$$

Since each column of $A$ has exactly one $1$ and the rest are $0$, we obtain that $u_i \in \{v_1, v_2, ...v_M\}$ for each $i = 1, 2, ..., N$. Let $e_i = (0, 0, .., 1, ...0)^\top$ be the unit vector in $\mathbb{R}^N$ that has a value $1$ only at $i$-th position and $0$ elsewhere. Let $W = Y_0^\top Y_0$ and $w_i$ be the $i$-th column of $W$. Notice that:

$$\begin{aligned} Y_0((BA - I_N)QF)[i] &= ||Y_0(u_i - e_i)||_F^2 \\ &= Tr((u_i - e_i)^\top Y_0^\top Y_0(u_i - e_i)) \\ &= (u_i - e_i)^\top W(u_i - e_i) \end{aligned}$$

So the original function can be simplified as:

$$\sum_{i=1}^{N} f_i(u_i - e_i)^\top W(u_i - e_i)$$

Now, let $C_i$ be the index set of those $j$ which satisfies $u_j = v_i$, which is the index set of a single cluster. Then the equation above can be considered independently on each $C_i$:

$$\begin{aligned} \sum_{i=1}^{N} f_i(u_i - e_i)^\top W(u_i - e_i) &= \sum_{i=1}^{M} \sum_{j \in C_i} f_j(v_i - e_j)^\top W(v_i - e_j) \\ &= \sum_{i=1}^{M} \sum_{j \in C_i} f_j(v_i^\top W v_i - e_j^\top W v_i - v_i^\top W e_j + e_j^\top W e_j) \\ &= \sum_{i=1}^{M} \sum_{j \in C_i} f_j(v_i^\top W v_i - 2w_j v_i) + \sum_{i=1}^{N} f_i e_i^\top W e_i \end{aligned}$$

Let $F_i = \sum\limits_{j \in C_i} f_j(v_i^\top W v_i - 2w_j v_i)$. This is a quadratic function for each $v_i$. Since $A$ has already been fixed, we know that $C_i$ is fixed. Thus we just need to optimize $F_i$ in each cluster.

Since $v_j$ can only have values on the indices of its corresponding cluster $C_i$, and all other positions must be 0, we have:

$$v_i = \sum_{j \in C_i} a_j e_j$$

Denote the element in the $i$-th row and $j$-th column of $W$ as $w_{ij}$. Thus we have:

$$F_i = (\sum_{j \in C_i} f_j)(\sum_{j \in C_i} a_j e_j)^\top W (\sum_{j \in C_i} a_j e_j) - 2\sum_{j \in C_i} f_j w_j (\sum_{j \in C_i} a_j e_j)$$

$$= (\sum_{j \in C_i} f_j) \sum_{j,k \in C_i} a_j a_k w_{jk} - 2\sum_{j,k \in C_i} a_k f_j w_{jk}$$

this is a quadratic function for $a_j$ ($j \in C_i$). Let $S_i = \sum\limits_{j \in C_i} f_i$, compute the derivative of $F_i$:

$$\frac{\partial F_i}{\partial a_j} = 2S_i \sum_{k \in C_i} a_k w_{jk} - 2\sum_{k \in C_i} f_j w_{jk}$$

$$\frac{\partial^2 F_i}{\partial a_j a_k} = 2S_i w_{jk}$$

Let $C_i = \{i_1, i_2, ... i_{|C_i|}\}$. We claim that if the 1-st derivative with respect to $(a_{i_1}, a_{i_2}, ..., a_{i_{|C_i|}})$ equals 0, then $F_i$ reaches a minimal value in this coefficient setting. Since $F_i$ is a quadratic function, the 3-rd derivative of $F_i$ equals 0. Consider the Taylor series of $F_i$, we've already know that the 2-nd derivative of $F_i$ equals $2S_i W$, which is a quasi-positive definite matrix. Then let $v'$ be the root of the 1-st derivative, we have:

$$F_i(v) = F_i(v') + (v - v')^\top \times \frac{\partial F_i}{\partial v}|_{v'} + (v - v')^\top \times 4S_i W \times (v - v')$$

$$= F_i(v') + (v - v')^\top 4S_i W (v - v') \geq F_i(v')$$

Now, let $a_{i_j} = \frac{f_{i_j}}{S_i}$, the 1-st derivative of $F_i$ equals:

$$\frac{\partial F_i}{\partial a_j} = 2S_i \sum_{k \in C_i} a_k w_{jk} - 2\sum_{k \in C_i} f_j w_{jk}$$

$$= 2S_i \sum_{k \in C_i} \frac{f_k}{S_i} w_{jk} - 2\sum_{k \in C_i} f_i w_{jk} = 0$$

To sum up, we've found a global minimal point for each $F_i$, which means that

$$v_i[j] = \begin{cases} \frac{f_j}{\sum\limits_{k \in C_i} f_k}, & \text{if } j \in C_i \\ 0, & otherwise \end{cases}$$

$\square$

## D  Implementation Details

Similar to M-SMoE, when reducing the number of experts from $N$ to $M$, we maintain $N$ references of experts while letting them point to $M$ real experts. In that way, the matrix $A$ is implicit encoded. In addition, for the compression matrix $T_1$, we calculate it in the GPU memory with the least square method. To maximize the number of samples used while avoiding out-of-GPU-memory errors, we adopt the BFloat32 data type. We perform the compression layer by layer. For each layer, we use Torch hooks to obtain intermediate activations, perform the least square method and release the memory after computation. The merging process traverses the layers from back to front because merging the later layers does not affect the activations of the earlier layers.

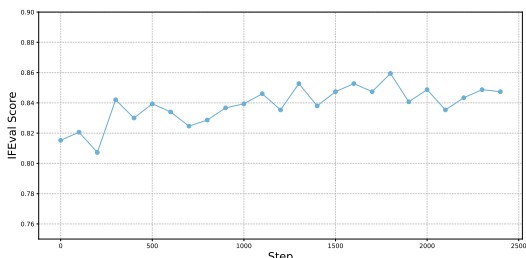

Figure 5: Evaluation on the IFEval benchmark.

# E  EXPERIMENTAL DETAILS AND ADDITIONAL EXPERIMENTS

## E.1  MODEL CONFIGURATIONS

In Table 6, we list their parameter size, the number of layers, the number of routed experts, the number of activated routed experts per token and whether they apply the shared experts architecture.

Table 6: Configurations for three used models in the evaluations.

| Model | Size | Layers | Experts | Activated Experts | Shared Experts |
|---|---|---|---|---|---|
| Qwen3-30B-A3B | 14B | 48 | 128 | 8 | No |
| Qwen1.5-MoE-A2.7B | 14B | 24 | 60 | 4 | Yes |
| DeepSeekMoE | 16B | 28 | 64 | 6 | Yes |

## E.2  HYPER-PARAMETER CONFIGURATIONS

We describe the hyper parameters in the comparative experiments. For the MergeMoE, when computing the compression matrix $T_1$ with the least square method, we conduct the computation in the GPU memory, and therefore the number of input samples used in the merging algorithm is limited. Besides, lengths of texts in different datasets may change, and therefore the batch size is also not fixed. In the comparative experiments we try to use large batch size for each dataset. We will ensure that, the batch size is the same for all merging algorithms applied to the same model and dataset combination.

**Comparative experiments on the Qwen3 model.**  For all merging algorithms, we merges the layers 28 to 47, reducing the number of experts in each layers from 128 to 64. For the number of input samples, we use 16 for ARC chanllenge, HellaSwag, PIQA, SQuAD, and 40 for the rest tasks.

**Comparative experiments on the Qwen1.5 model.**  For all merging algorithms, we merges the layers 10 to 23, reducing the number of experts in each layers from 60 to 30. For the number of input samples, we use 32 for PIQA and SQuAD, and 64 for the rest tasks.

**Comparative experiments on the DeepSeekMoE model.**  For all merging algorithms, we merges the layers 16 to 27, reducing the number of experts in each layers from 64 to 28. For the number of input samples, we use 128 for WinoGrande and MRPC, 64 for ARC easy, ARC challenge and Hellaswag, and 40 for the rest tasks.

## E.3  EVALUATION ON IFEVAL

We further evaluate our algorithm on the IFEval benchmark. The evaluation is conducted on the Qwen3-30B-A3B, and we use the same compression configuration as in Appendix E.2, which reduces the number of model parameters from 30B to 25B. We additionally incorporate ShareGPT for knowledge distillation, aiming to explore whether instruction-following ability could be further

enhanced. As shown in Figure 5, without any distillation, the compressed model achieves a score of $0.8153$. With knowledge distillation, its performance is further boosted to around $0.85$. This demonstrates two key findings: our merging algorithm yields solid results even in its compressed form, and knowledge distillation can serve as an effective means to further enhance performance on generative tasks.

