# OpenReview forum: "MergeMoE: Efficient Compression of MoE Models via Expert Output Merging"
_ICLR.cc/2026/Conference — ICLR 2026 Conference Withdrawn Submission_

### Official Review · Reviewer_Dpo1 · 2025-10-20

**Soundness:** 3
**Presentation:** 3
**Contribution:** 2
**Rating:** 4
**Confidence:** 3

**Summary:**

This paper addresses the substantial memory overhead issue of Mixture-of-Experts (MoE) models by proposing MergeMoE, an efficient compression method. Unlike conventional expert merging techniques that approach the problem from parameter aggregation perspective, the authors innovatively reframe expert merging from the viewpoint of merging expert outputs, reinterpreting the merging process as inserting additional matrices into forward computation, which naturally leads to a mathematical optimization formulation. Specifically, the algorithm first clusters experts based on the similarity of their weight matrices, then leverages mathematical optimization to construct the merged experts.

**Strengths:**

- The authors provide comprehensive theoretical proofs and mathematical derivations for their proposed merging theory. The method is grounded in solid mathematical foundations.
- The paper demonstrates relatively clear writing that makes the authors' insights and concepts more accessible and easier to understand.
- The authors appropriately provide model size after compression and computational cost analysis in the experimental section, which enables better evaluation of the practical utility and efficiency of the proposed compression technique.

**Weaknesses:**

- The paper only evaluates on Qwen and DeepSeek model families, making it unclear whether the conclusions would hold for other MoE model architectures (e.g., GPT-OSS), which raises questions about the generalizability of the proposed method across different model designs and architectures.
- The paper only evaluates on language tasks, lacking evaluation on important domains such as mathematics and code generation tasks, which limits the assessment of the method's effectiveness across the full spectrum of capabilities that modern MoE models are expected to handle.
- The paper lacks discussion of current quantization techniques, such as the performance when quantization is applied alongside expert merging, despite quantization being widely used in practice and orthogonal to model compression approaches.
- MergeMoE's similarity-based clustering approach and frequency-based merging strategy are similar to prior works (e.g., [1,2]), suggesting that the novelty is somewhat incremental.

[1] Merge, Then Compress: Demystify Efficient SMoE with Hints from Its Routing Policy

[2] Retraining-Free Merging of Sparse MoE via Hierarchical Clustering

**Questions:**

1. What would be the performance of MergeMoE on other MoE architectures beyond Qwen and DeepSeek families?
2. What is the performance of the compressed models on other tasks?
3. What is the performance when MergeMoE is combined with quantization techniques?

---

### Official Review · Reviewer_Hg2e · 2025-10-28

**Soundness:** 3
**Presentation:** 3
**Contribution:** 2
**Rating:** 2
**Confidence:** 4

**Summary:**

The paper revisits expert merging for MoE models from an **output-space** perspective rather than parameter averaging. It formalizes merging as inserting linear maps into the forward pass and proposes a framework with clustering/weighting ((A,B)) and three projection matrices $((T_1,T_2,T_3))$. In practice, the method fixes $(T_2,T_3)$ as **frequency-based merges** within clusters and then computes $(T_1)$ via a closed-form least-squares solution $(T_1 = QP^{\dagger})$. Experiments on several MoE models report improvements over prior merging baselines at comparable compression ratios.

**Strengths:**

**1. Originality:** The output-merging view is conceptually clear and helps unify and reinterpret prior parameter-space approaches (e.g., showing them as special cases).

**2. Quality:** The framework is well presented; the role of $(A,B,T_1,T_2,T_3)$ in approximating the original forward pass is easy to follow.
Practicality: Computing $(T_1)$ with a least-squares closed form is simple and efficient, making the procedure easy to reproduce.

**3. Significance:** If broadly applicable, the output-space perspective could influence how MoE compression methods are designed beyond naive weight averaging.

**Weaknesses:**

**1. Core conclusions hinge on a strong assumption:**

All theory and performance gains rely on **fixing $(T_2,T_3)$ to frequency-merge** (cluster-wise usage-frequency weighting). The “optimality” claims are therefore **conditional**—they hold under this parameterization (and additional assumptions such as independence between router logits and expert outputs), not for more general or learnable $(T_2,T_3)$. This limits generality and reduces the paper’s contribution to “optimal within a restricted family.”

**2. Design space underexplored:**

There is no systematic comparison against alternative choices for $(T_2,T_3)$ (TIES, Fisher, DARE merge in model merging methods), nor against other weighting schemes beyond frequency.

**3. Operational metrics missing:**

Inference-time effects (throughput, latency, memory) after merging are not reported, making practical value harder to assess beyond parameter counts.

**Questions:**

My only major consideration is this: the key results and optimality are conditional on $(T_2,T_3)$ being frequency merges. Demonstrating effectiveness with more general $(T_2,T_3)$—or providing strong evidence that frequency weighting is uniquely preferable—would significantly strengthen the contribution.

**Details Of Ethics Concerns:**

NO or VERY MINOR ethics concerns only

---

### Official Review · Reviewer_DVn9 · 2025-10-29

**Soundness:** 2
**Presentation:** 2
**Contribution:** 1
**Rating:** 2
**Confidence:** 5

**Summary:**

This paper introduces MergeMoE, a post-training compression approach for Mixture of Experts (MoE) models. The method identifies redundant experts through cosine similarity of their output activations and merges similar experts using low-rank approximation with singular value decomposition. A small residual MLP is optionally added to compensate for the approximation error. The authors evaluate the method on several public MoE architectures such as OLMoE and DeepSeek-V2-Lite and report moderate compression with limited degradation on general understanding benchmarks.

**Strengths:**

- The motivation of reducing redundancy among experts is reasonable and supported by observations from over-parameterized MoE models.
- The proposed merging scheme is simple and easy to integrate into existing systems, which could facilitate efficient deployment without retraining.
- Experiments are conducted on multiple MoE backbones and datasets, providing a relatively broad empirical view.

**Weaknesses:**

- The core idea of expert merging has been explored in prior literature. Although this work uses singular value decomposition for merging, the contribution beyond existing methods remains limited.
- The reported results show noticeable performance degradation under stronger compression. The paper would benefit from a clearer analysis of when merging is beneficial and when it becomes detrimental.
- The evaluation focuses mainly on understanding-oriented classification tasks such as HellaSwag. It remains unclear how the proposed method performs on more generation-intensive or reasoning-based tasks, where MoE architectures may behave differently.
- The practical inference efficiency gains are not clearly analyzed, making it difficult to assess the real deployment advantages given the observed performance loss.

**Questions:**

- Since the method relies on post-hoc distillation to recover performance, it would be useful to separate the effects of merging and distillation.

- The paper could be improved by including more challenging benchmarks such as code generation, mathematical reasoning, or long-context modeling, which would make the study more relevant to large-scale model evaluation.

---

### Official Review · Reviewer_BGKX · 2025-11-01

**Soundness:** 3
**Presentation:** 2
**Contribution:** 3
**Rating:** 4
**Confidence:** 4

**Summary:**

The authors present MergeMoE, a compression technique for Mixture-of-Experts (MoE) models. The work reframes expert merging from the conventional parameter-aggregation view to an output-merging perspective. This insight formulates compression as a quantitative optimization problem: finding matrices ($A, B, T_1, T_2, T_3$) that minimize the output approximation error. The method uses a decoupled optimization strategy, setting $T_2$ and $T_3$ via weighted averaging and solving for $T_1$ analytically using a least-squares solution on sampled data. The authors also provide theoretical justification (Theorem 1) for using expert usage frequency as the optimal weighting scheme.

**Strengths:**

- The reformulation of expert merging as an output-space optimization problem is a strong theoretical contribution, moving beyond prior heuristic-based parameter-space approaches
- The paper provides a novel theoretical proof (Theorem 1) justifying the optimality of using usage frequency for weighting, a method previously adopted by M-SMOE based only on empirical performance
- The proposed method, particularly the use of least-squares to solve for the $T_1$ compression matrix, demonstrates consistently superior performance over baselines (M-SMOE, ZipIt) across multiple MoE architectures and benchmarks

**Weaknesses:**

- The optimization is decoupled and incomplete. The method defaults to a simple heuristic (weighted averaging) to set $T_2$ and $T_3$, which the paper notes is identical to the M-SMOE baseline's approach. This weakens the claim of a fully optimization-based approach, as the non-linear components are not optimized
- The main experimental evaluation (Tables 1-3) is weak, as it presents only a single, fixed compression ratio for each model (e.g., 128->64 for Qwen3, 60->30 for Qwen1.5, 64->28 for DeepSeek). This makes it impossible to assess how the methods compare at different, more aggressive compression levels (e.g., 4x, 8x)
- The ablation study on compression ratio impacts (Figure 2) is insufficient. It is limited to a single model (Qwen 1.5) and a single dataset (WinoGrande), preventing generalizable conclusions about the performance/compression trade-off
- The method's performance is shown to be highly sensitive to the number of input samples, failing completely (random-guess performance) below a "critical threshold". This sensitivity (Figure 4) is a significant practical limitation, and its dependency on sample quality or distribution is not explored
- The comparison is limited to other expert-merging techniques. The lack of comparison against other established MoE compression methods (e.g., expert pruning based on usage frequency, or quantization) makes it difficult to contextualize the method's practical utility

**Questions:**

- The optimization of $T_2$ and $T_3$ is avoided. What is the performance impact of learning these matrices, for instance, via gradient descent on the least-squares objective, rather than using the M-SMOE heuristic?
- Can the authors provide benchmark comparisons (vs. M-SMOE, etc.) at more aggressive compression ratios (e.g., 4x, 8x expert reduction) in Tables 1-3?
- Given the method's high sensitivity to sample size (Figure 4), how stable is the result based on the distribution of the (e.g.,) 36+ samples? Does a poorly chosen batch of samples lead to catastrophic failure?
- How does MergeMoE compare to a simple expert-pruning baseline that removes the M-N least-frequently-used experts and fine-tunes the router?

---

### Note · Authors · 2025-11-15

I have read and agree with the venue's withdrawal policy on behalf of myself and my co-authors.